# Exploring EEG spectral and temporal dynamics underlying a hand grasp movement

**Sandeep Bodda**[1], **Shyam Diwakar**[1,2]*

**1** Amrita Mind Brain Center, Amrita Vishwa Vidyapeetham, Kollam, Kerala, India, **2** Department of Electronics and Communication Engineering, School of Engineering, Amrita Vishwa Vidyapeetham, Kollam, Kerala, India

* shyam@amrita.edu

**Data Availability Statement:** Data collected for this research contains physiological information of particular subjects. There are ethical restrictions on sharing the data set. The consent given by participants at the outset of this study did not explicitly detail sharing of the data in any format;

## Abstract

For brain-computer interfaces, resolving the differences between pre-movement and movement requires decoding neural ensemble activity in the motor cortex's functional regions and behavioural patterns. Here, we explored the underlying neural activity and mechanisms concerning a grasped motor task by recording electroencephalography (EEG) signals during the execution of hand movements in healthy subjects. The grasped movement included different tasks; reaching the target, grasping the target, lifting the object upwards, and moving the object in the left or right directions. 163 trials of EEG data were acquired from 30 healthy participants who performed the grasped movement tasks. Rhythmic EEG activity was analysed during the premovement (alert task) condition and compared against grasped movement tasks while the arm was moved towards the left or right directions. The short positive to negative deflection that initiated around -0.5ms as a wave before the onset of movement cue can be used as a potential biomarker to differentiate movement initiation and movement. A rebound increment of 14% of beta oscillations and 26% gamma oscillations in the central regions was observed and could be used to distinguish pre-movement and grasped movement tasks. Comparing movement initiation to grasp showed a decrease of 10% in beta oscillations and 13% in gamma oscillations, and there was a rebound increment 4% beta and 3% gamma from grasp to grasped movement. We also investigated the combination MRCPs and spectral estimates of α, β, and γ oscillations as features for machine learning classifiers that could categorize movement conditions. Support vector machines with 3rd order polynomial kernel yielded 70% accuracy. Pruning the ranked features to 5 leaf nodes reduced the error rate by 16%. For decoding grasped movement and in the context of BCI applications, this study identifies potential biomarkers, including the spatio-temporal characteristics of MRCPs, spectral information, and choice of classifiers for optimally distinguishing initiation and grasped movement.

## 1. Introduction

Patients with amyotrophic lateral sclerosis (ALS) or spinal cord injury (SCI) are reported to have significant loss of voluntary motor control and extensive dysfunction of upper and lower

this limitation is imposed by the Research Ethics Committees of the university and hence cannot be shared. For this reason, data collected is not currently available in a public repository. However, all the data is available upon request. Please contact the committee head & Dean, life sciences from the university, Dr. Bipin Nair. contact: bipin@amrita.edu, mindbrain@amrita.edu.

**Funding:** This study was partially supported by the Department of Science and Technology Grant DST/CSRI/2017/31, Government of India and Embracing the World research for a cause Initiative.

**Competing interests:** The authors have declared that no competing interests exist.

limbs [1–4]. Rehabilitation using robotic or functional electrical stimulation could facilitate novel therapeutic strategies for such conditions. Integration of signals could also be a potential goal for Brain-Computer Interfaces (BCI), allowing the transfer of control information from a prosthesis onto the brain tissue [5–7].

Neural correlates of voluntary grasped movement are relevant in the development of modern prosthetics and towards Brain-Computer Interface (BCI) research [8–10]. Insights from electroencephalography (EEG) recordings and their underlying neurophysiological processes involved in motor tasks can help decode movement and its functional interpretation. EEG signal components related to movement allow noninvasive measurements, making them suitable for natural BMIs [5, 6]. In addition, these correlates are measurable in paralyzed patients [3, 4, 7, 11]. For example, the assessment of grip forces [12], movement direction, and lifting paradigms [13] have allowed to objectively and quantitatively detect deficits in patients with Multiple Sclerosis (MS). Studies [14–16] have also focused on low-frequency EEG components to understand how movement trajectories [17] and grasp types [18] were encoded.

EEG-based studies have explored movement initiation or intention and execution to be associated with μ/α and β rhythms [19], although the role of higher frequency oscillations in movement execution remains to be critically explored. The modulation of β rhythms and oscillations in the context of sensorimotor state maintenance and motor function has been previously reported [20, 21]. However, there is little information related to cortical zone-dependent microstates and their movement-related cortical potential (MRCP) components and their variability during resting states for reach, grasp, and grasped movement tasks performed using a human arm. Readiness potentials, and time delays to occurrences of motor-related events while simultaneously comparing motor imagery and motor execution spectral components in EEG can extend current knowledge for developing BCIs [10]. Event-related (de)synchronisation ERD/S and movement-related cortical potentials (MRCPs) have been employed to assess and detect movement intention, execution and imagery in several studies [22–26]. Among the neural correlates of voluntary movement intention, studies had reported the role of a component of MRCP, the slow negative potential [27, 28]. MRCPs have been observed as low-frequency components within EEG signals, computed by averaging several trials commencing 2s before voluntary movement [29, 30]. The readiness potential (RP) [31], an MRCP component, was perceived with a larger amplitude at contralateral central regions around 400ms before the movement onset. In the case of conscious voluntary movement, readiness potential was known to precede 500-800ms before the onset of movement [32, 33]. MRCPs serve as reference attributed to cortical excitability [34–36] occurring before the movement. Also, MRCPs have been used as biomarkers of movement intention and movement tasks in ALS [37], stroke [25], and Parkinson's disease [38, 39] patients.

The neural components correlated to hand-reaching that occurred within the medial parieto-frontal circuit including the medial intraparietal area (mIP) at the boundaries with area V6A and the dorsal pre-motor areas have been explored [40]. Neural activity related to grasping was reported to occur in the lateral parieto-frontal circuit involving the anterior intraparietal area (AIP) primarily and the dorsal (PMd) and the ventral (PMv) regions of pre-motor areas [41–43]. A significant role was known to be identified in the human AIP (hAIP) during grasping tasks [44–49]. It was demonstrated [50] that grasp-related neural activity was observed in the ventral premotor F5, parietal AIP, the cortical F1 areas pertaining to the hand field and in the several regions of the somatosensory cortex. An intrinsic spatial optical imaging (ISOI) study [51] had explored spatial domains that were active in M1 and S1 in response to reaching and grasping in macaque monkeys.

Assessing synchrony and desynchrony of rhythmic oscillations had allowed dissecting cortical preparation or execution of voluntary movements [52–54]. In the motor cortex, μ

rhythms in the α band [22, 55], and attenuation of the α-band were observed during the preparation or execution of voluntary movements, which was also accompanied by a decrease in the β band decrease, known as event-related desynchronization (ERD). Further, a rebound of the β-band after the movement, known as event-related synchronization (ERS) [22] was used to detect changes in movement execution. From an application standpoint, ERD/ERS features allow the reuse EEG signal information within the context of a brain-machine interface (BMI) and similar methods in the anthropometric articulation of external devices [53, 54, 56–58]. Control paradigms were employed to operate wheelchairs using decoded ERD/ERS [59, 60] information. During the movement preparation and execution, ERD was explored for investigating cortical modifications and was shown the delayed ERD in Parkinson's Disease (PD) patients compared with healthy subjects over the contralateral sensorimotor areas [22, 61]. In the S1 area, for self-paced finger or wrist-extension movement, a decrease in 10–14 Hz frequencies and a transient increase in activity (ERS) after the movement for 20 and 90 Hz frequencies were observed [62]. An increase in the 40 Hz gamma-band activity (ERS) over the central scalp region was reported during both simple response time (SRT) and complex response time (CRT) task reactions [63, 64]. Studies show correlations in the slow (6–11 Hz) and to the higher © band region of (60–120 Hz) [65, 66] corresponding to any of the 4 fingers being moved and their flexion and trajectories of finger movements during grasping.

Other than μ or β band activity, studies have reported γ band activity in the motor and premotor areas from intracranial electrode recordings and electrocorticogram recordings [67–69]. The subdural recordings of electrocorticography (ECoG) signals performing visuomotor tasks designed to activate representations of different body parts had shown © bands were distinct in the sensorimotor cortex and the ERS of γthe band has been observed during or just before the motor response and lasted for a short time and ended before the completion of the motor response. The temporal differences associated with the γ bands had implicated functional associations with motor performance [64]. During the execution phases of movement, γ band synchronisation was also reported, and γ clusters contralateral to movement have been observed [70, 71]. The findings [70] of Ramos-Murguialday and colleagues had indicated that active movement could be decoded using a low γ-band in EEG after filtering EMG artifacts from channel data.

There has been an increased demand for wearable technologies and especially, limited-channel systems for personal use. Comparing clinically graded systems and consumer systems and the applications [72–74] have been reported. The application of these consumer systems in research has not been extensively explored. It was also crucial to explore whether a limited-channel acquisition could provide interpretable data to assess the neural signatures while exploring grasped movement.

In other BCI studies [75–80], various grasp movements have been used in actual and imagined motor tasks but due to task complexity, they have not been interpreted for BCI and related neurophysiological studies. The MRCPs and spectral oscillations relationship have not been investigated in detail and remain uncertain, although both phenomena accompany sensorimotor activity, and their underlying neural circuits have been hypothesized to reside within the same anatomical structures. A few studies have explored their relationship from a neurophysiology perspective [81–84]. Through this study, the neural correlates could provide complementary information on associated grasped movement tasks where MRCPs could be related to cortical excitability shifts and ERD/ERS to the gating of task-specific thalamocortical circuits [85]. In the context of early movement prediction and detection, the number of studies addressing the relation of MRCP and ERD/ERS morphology and the classification of such data are limited [2, 24, 25].

Since lower resolution EEG was less expensive and easier to implement than the other temporally precise methods, and with the limits of EEG spatial resolution not well understood, whether specific EEG electrodes are sufficient for movement-related spatial information remains unanswered. This study explores the EEG dynamics by employing a novel protocol to understand movement intention and execution during grasped movement tasks performed in the left and right directions. While exploring the activity space underlying grasping controlled by parieto-frontal circuits [86] and central regions, this paper also attempts to relate the variations in movement initiation and grasped movement by developing classification models based on features of temporal and spectral correlations specifically with a combination of α, β, and lower γ frequency bands and readiness potentials for grasped-movement.

In our study, the rebound activity of β and γ oscillations after the movement and the slow negative deflection amplitude shift in the temporal domain were explored as these features could be potential cortical activity biomarkers for grasped movement. In this paper, we also evaluated feature-based classification of pre-movement and grasped movement using machine learning and have enumerated ranked features from these tasks that may be relevant for constructing efficient classifiers for clinically relevant movement EEG data.

## 2. Methods

Thirty healthy volunteers aged 18 to 30 years (mean age = 22.32 ± 1.92 years) participated in this study. This non-invasive study was reviewed and approved by the institutional ethics committee at the university. All subjects involved in this study were without any known prior medical conditions and with normal or corrected-to-normal vision. All subjects were explained the aim of the study and signed informed consent were collected before their participations in the recordings. The tasks included were grasped movement of bottle towards using left hand towards the left direction and with right-hand towards the right direction.

### 2.1 Experimental paradigm

The experimental setup and the task paradigm were adapted from [16] (see Fig 1). A cue-based paradigm (Fig 1B) was employed where subjects were presented with visual cues using a slide-based presentation for 45 seconds. During the recordings, the subjects were seated in a comfortable chair and the computer display was at eye level and an object to grasp was placed at a random distance of 69–82 cm (approximately the length of reaching the object by the subject's arm) from the subject (Fig 1A). Subjects were asked to place their left or right arm parallel to the table while resting on the hand-rest of the chair. The object, a 6.5 cm diameter rigid bottle was positioned on the centre of the table.

Prior to the experiment, participants were provided training to adapt to the task by progressive steps of hand movement by grasping the object. Before recordings, the subjects repeated the same procedure three or four times until they attained familiarity to perform the task.

The experimental paradigm was conducted as outlined in the following steps.

1. All trials commenced with a relaxation phase (blank screen), considered as a reference or baseline signal for the analysis

2. The subject was asked to relax for ten seconds.

3. The subject was alerted for the following task by a '+' sign cue for 5 seconds.

4. The word 'Reach' as a cue was shown for 5 seconds, indicating the subject to reach the object.

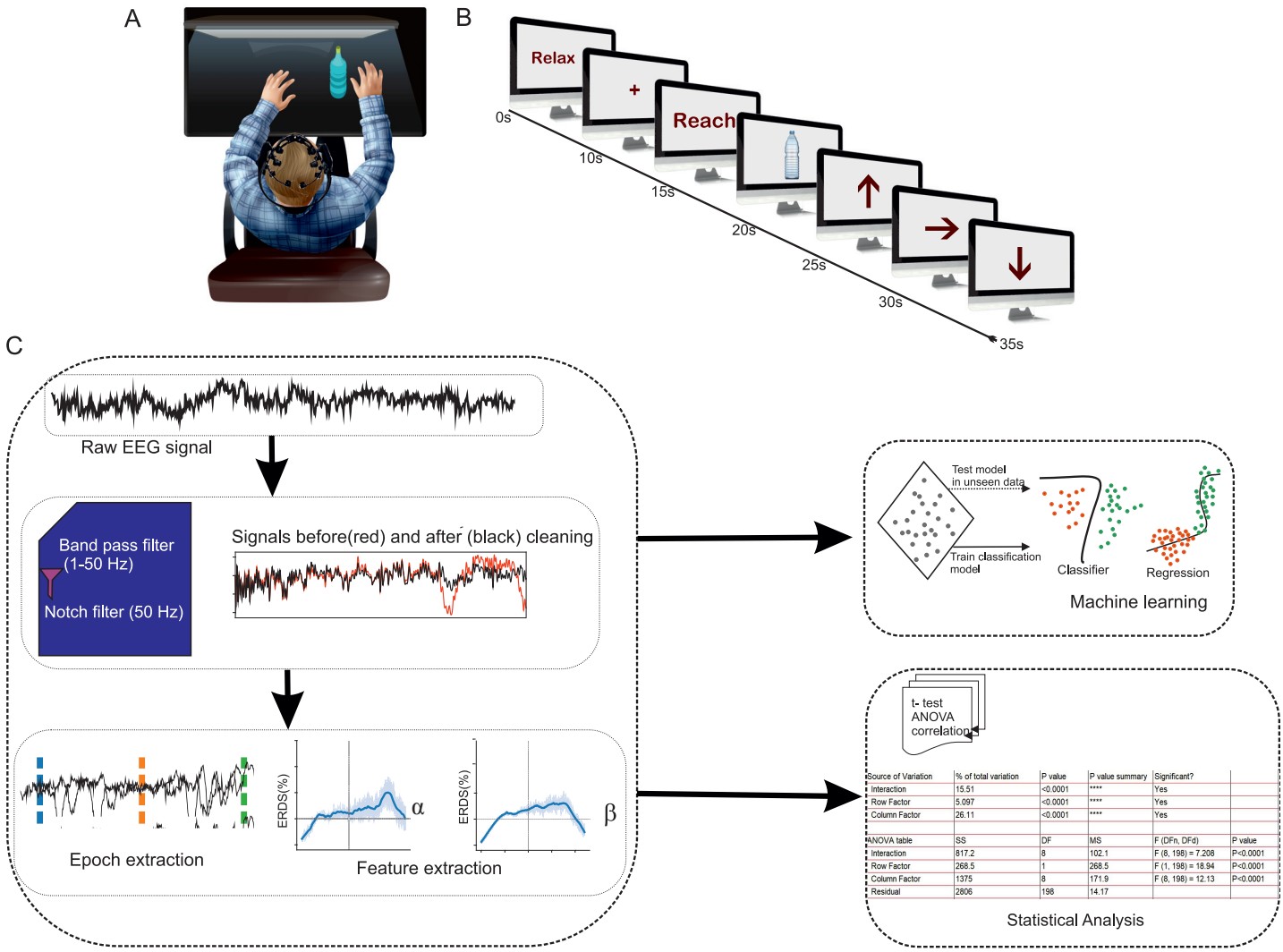

**Fig 1. Schematic representation of experimental protocol and configuration.** (A) Sitting posture of the subject during the neurophysiological recording (B) The timeline points to the temporal reference for various tasks initiated with a 'relax' state that started at 0s, "+" visual cue at 10s for the subject to be 'alert', at 15s ' Reach' cue indicated the subject to reach for the object placed in the front of the subject, and at 20s 'Object' cue appeared to indicate the subject to grasp the object, followed by "arrow" cue at 25s to lift the bottle, consequently followed by another 'arrow' cue at 30s that suggested the direction in which the object was to be moved and a 'down arrow' cue at 35s that suggested to place the grasped object back at rest. (C) EEG Data processing analysis workflow involved the filtering of raw signal, epoch extraction, feature extraction and building a machine learning model based on processed features.

5. Then an image of a bottle was shown to the subject, indicating to grasp the bottle placed in front of the table. This cue was presented for five seconds

6. An upward arrow cue (↑) was shown for next 5 seconds indicating to the subject to lift the bottle to the chin level.

7. This was followed by a leftwards arrow cue (←) for five seconds; subject was then instructed to move the bottle towards left direction using the left hand.

8. Following this, a down-wards arrow cue (↓) was shown for 5 seconds indicating to the subject to place the object back down on the table

9. A blank screen was shown as an indication of the end of the experiment trial

The same steps were repeated for right-hand right direction movement task as well.

The tasks were carried out using both hands for the two movement directions in trials defined as right-hand leftward direction (RHLD), right-hand rightward direction (RHRD), and left-hand leftward direction (LHLD), and left-hand rightward direction (LHRD). However, only LHLD and RHRD have been considered for the analysis.

## 2.2 EEG data acquisition

The study included the acquisition of EEG data by placing EEG sensors on the subject's scalp. For clinically relevant data, we used a 32-electrode commercially available device (Neuroelectrics, Barcelona, Spain) positioned on the scalp according to the 10–20 international system with a sampling rate of 500Hz.

Considering commercial and limited electrode platforms, we also employed a 14-electrode device (EMOTIV EPOC+) with a sampling rate of 128 Hz. EEG signals were recorded for 10 subjects using a 32- electrode device. For each subject, four trials were performed per task. So, a total of 40 trials were performed for 10 subjects. Due to the loss of data packets, EEG data of 12 trials were discarded. Using the 14-electrode device, EEG signals were recorded for 20 subjects. In this, 10 trials were performed for 6 subjects, 5 trials for 12 subjects, 13 trials for 1 subject, and 2 trials for 1 subject have been carried out and EEG signals were recorded. Out of 175 trials, 163 were used for the data analysis, and 12 trials were rejected due to the loss of signal packets during the recording.

## 2.3 Offline processing of EEG data

Data analysis was performed using custom scripts written in Python incorporating the 'mne' package functions [87]. To obtain the spectral components initially the data was high pass filtered (1Hz) to minimise the drifts [88] and the reference (mean) was subtracted, further the data was detrended [89] and bandpass filtered using an FIR filter [90, 91] of order 20 within the range, 1Hz—60 Hz and notch filter was applied to remove line noise in the range of 50 Hz. Independent Component Analysis (ICA) [92, 93] was used to eliminate the EOG, EMG artifacts from the data. ICA considered n number of linear mixtures $X_1$, $X_2$. . . . . .., $Xn$, n number of components in this case total number of channels have considered n = 32 number of components. signal X:

$$X = As \tag{1}$$

*A* represents mixing matrix with size n×n and s was the vector of independent components. The mixing effect, after computing the matrix generated the independent components

$$y = wX \cong S \tag{2}$$

To obtain the independent components in this study, we used the infomax algorithm based on general optimisation principle for neural networks and other processing systems [93, 94]. The algorithm determined weights based on the maximation of output entropy of a neural network with nonlinear outputs:

$$w_{k+1} = w_k + \mu_k \left[ I - 2g(y_k)y_k^T \right] w_k \tag{3}$$

where: y–matrix of source estimation (y = Wx); k–number of iterations; I–the identity matrix; $\mu_k$–learning rate which depended on k; g (.)–a nonlinear function. *g(y)* logistic function.

$$g(y) = \frac{1}{1 + e^{-y}} \tag{4}$$

Further, data was segmented into non-overlapping epochs of 2s for the timelines of 0s - 10s (Relax Task), 10s-15s (premovement task), and 30s -35s (Left hand left direction movement / Right hand right direction movement). Consequently, the segmented data was used to estimate the spectral features. The spectral estimations of each rhythm were quantified using multitaper power spectral density (PSD) estimation [95, 96]. The standard multitaper PSD [97] consisted of a series of steps: multiplying each data segment by each taper, applying Fourier analysis to these products, averaging over the tapers within each segment, and averaging over the segments. The PSD was estimated for the frequency ranges of δ, θ, α/μ, β, and γ bands. The global field power (standard deviation) across the regions (central, frontal and parietal) of electrodes was obtained.

For temporal cortical potential components [29], The MRCPs typically occur at frequencies of around 0–5 Hz [98]. To minimise the drifts from the raw data reference (mean) was subtracted, further the data was detrended [89] and bandpass filtered using an FIR filter [90, 91] of order 20 within the range of 0.1Hz- 5 Hz. Further, Multiple recordings of the same trials must be taken and then averaged across the trials for extraction of MRCP from EEG traces. By averaging, the background noise is cancelled out leaving only the MRCPs, when the data from multiple trials is filtered to eliminate the higher frequency activity.

## 2.4 Statistical measures

Correlation analysis was carried out to determine the best electrode positions for the analysis of MRCPs. The data from channels (C3, C4, F3, F4, P3, P4) were selected and used for correlation analysis. The relation for electrode positions in different regions of cortical regions from MRCPs was analysed using Pearson correlation coefficient. For each electrode channel and N scalar observations (time points), the correlation coefficient was computed.

$$\rho(x, y) \ = \ \frac{1}{(N-1)} \sum_{i=1}^{N} \left( \frac{\overline{x_i - \mu_x}}{\sigma_x} \right) \left( \frac{\overline{y_i - \mu_y}}{\sigma_y} \right) \tag{5}$$

Where x and y were two variables (electrode channels) corresponding to two electrode positions under analysis. The $\mu_x$ and $\sigma_x$ values were mean and standard deviation of x (channel 1) and the $\mu_y$ and $\sigma_y$ were the mean and standard deviation of y (channel 2) respectively. The goal of the correlation analysis was to identify the optimal electrode position for the relax, premovement and movement tasks. Further, the correlated channels were compared to test the significance of correlated channels for each task using multiple t-test statistics. Statistical significance was determined using the Holm-Sidak method, with alpha = 0.05. Each row (correlated pair combination of electrode) was analysed individually, without assuming a consistent SD.

The association between frequency band oscillations in the three tasks and the subject's gender was tested using $\chi^2$ analysis. The value of $\chi^2$ was estimated using the formula,

$$\chi^2 = \frac{(F_O - F_E)^2}{F_E} \tag{6}$$

where $F_O$ was the observed frequency count and $F_E$ was expected frequency count of dependent features, the confidence interval chosen was 0.05. Similar analysis was performed to find the association between frequency band and tasks (the average count of the frequency bands was used in the contingency table for analysis).

To better understand the relationship between changes in PSD for central electrode regions across different frequency bands for all 28 trials, PSD for tasks (relax, premovement, movement) was compared. A two-way repeated measure of ANOVA with Tukey's multiple

comparison posthoc test was performed on the average of each feature extracted from the relax, pre-movement and grasped movement conditions. Similarly, we compared the frequency band features for the task's male and female subjects for all the trials, to identify the statistical significance of the features discriminating among the male and female subjects for each task using two-way ANOVA analysis. All the statistical analysis was performed using Graph-Pad Prism [99].

## 2.5 Machine learning

Using Decision trees (DT) [100–102], support vector machines [103, 104] with different kernels, and multilayer perceptron [105, 106] with different activation functions and solvers, classification was performed on the feature-combined dataset including both spectral and temporal (RPs) information for the premovement vs movement tasks and left-hand grasped-movement vs right hand grasped movement tasks. Here, DT (criterion: '*Gini*', splitter: '*best*'), SVM model [107–109] (regularisation parameter C = 1.0, Kernel: linear, polynomial and radial basis function, degree = 3) and for MLP (activation function: logistic sigmoid function, quasi-Newton method as a solver: 'lbfgs', hidden_layer_sizes: (35,2), maximum number of iterations = 300). The implementation of the algorithms was based on the scikit-learn [110] python package and was chosen for the classification of grasped movement data.

Two EEG datasets consisting of 270 samples (135 samples premovement, 135 movement trials) and 56 samples (trials or instances) (28 premovement, 28 movement trials) respectively were used in this machine learning analysis. The two datasets contained 67 and 74 features as columns which were pre-processed values from the raw data. The classifier models were trained, and accuracy was estimated using 10-fold cross-validation and the datasets had split into training and testing samples with a random choice. To explore the differences in the accuracies of the datasets, Wilcoxon signed ranked t-test [111] was employed, and the t-test had indicated that there were no significant effects (p = 0.65) in the classification accuracies when using electrodes from two different headsets data (also see S4 Fig in S1 File).

**2.5.1. Pre-classification feature selection for pre-movement and grasped movement tasks.** 20 best-ranked features in the dataset were identified using the feature selection ranker search algorithm [112, 113], which indicated that the frequency sub-bands of μ/α (7–10 Hz), β (18, 25–29 Hz), γ (30–32 Hz) and related potential (see Fig 2).

## 3. Results

### 3.1 MRCP-related time-domain variations allow to decode grasped-movement

Associated to C3-C4 electrodes (see S1 Fig in S1 File), the postcentral, precentral gyrus and the primary motor cortex (M1) has a role in the initiation and fine control of movement; and, in this study, the central electrode regions (C4-C3) reported a higher correlation measure (r = 1) for movement and "alert"/pre-movement conditions compared to "relax" or no-movement state (r = 0.3) (Fig 3A, also see S4 Table in S1 File for other electrode regions). A combination of parieto-central and frontocentral region electrodes showed a correlation measure of 0.8 and 0.4 for movement and premovement conditions respectively (Fig 3A). Parieto-frontal (P4-F4) regions reported a correlation measure of 0.5 for the premovement/ "alert" and "relax" tasks, 0.6 for the "movement" task. The correlation-based analysis recommended C3-C4 combination for premovement and movement tasks due to the higher values compared to P4-C4, P4-F4, P4-C3, C4-F4, C4-F3 combinations.

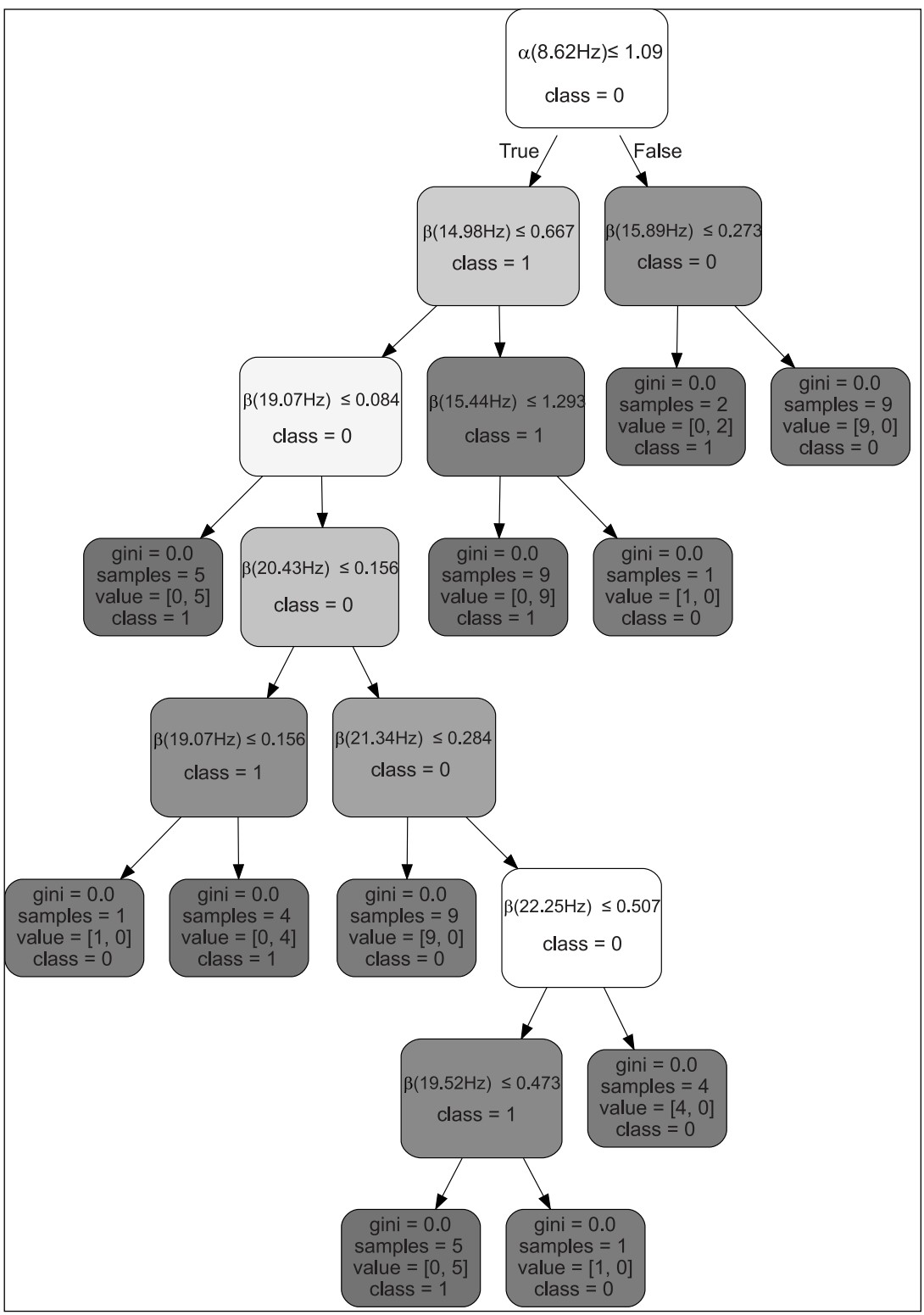

**Fig 2. Feature ranking of surface EEG signals from central electrodes C3 and C4 showed accuracy depended on α sub-band 8.62 Hz and β sub-bands 14, 15, 19, 20, 21, 22 Hz while classifying movement (gray) against premovement (light grey/ white) class labels (Class 0 is premovement, Class 1 is grasped movement).**

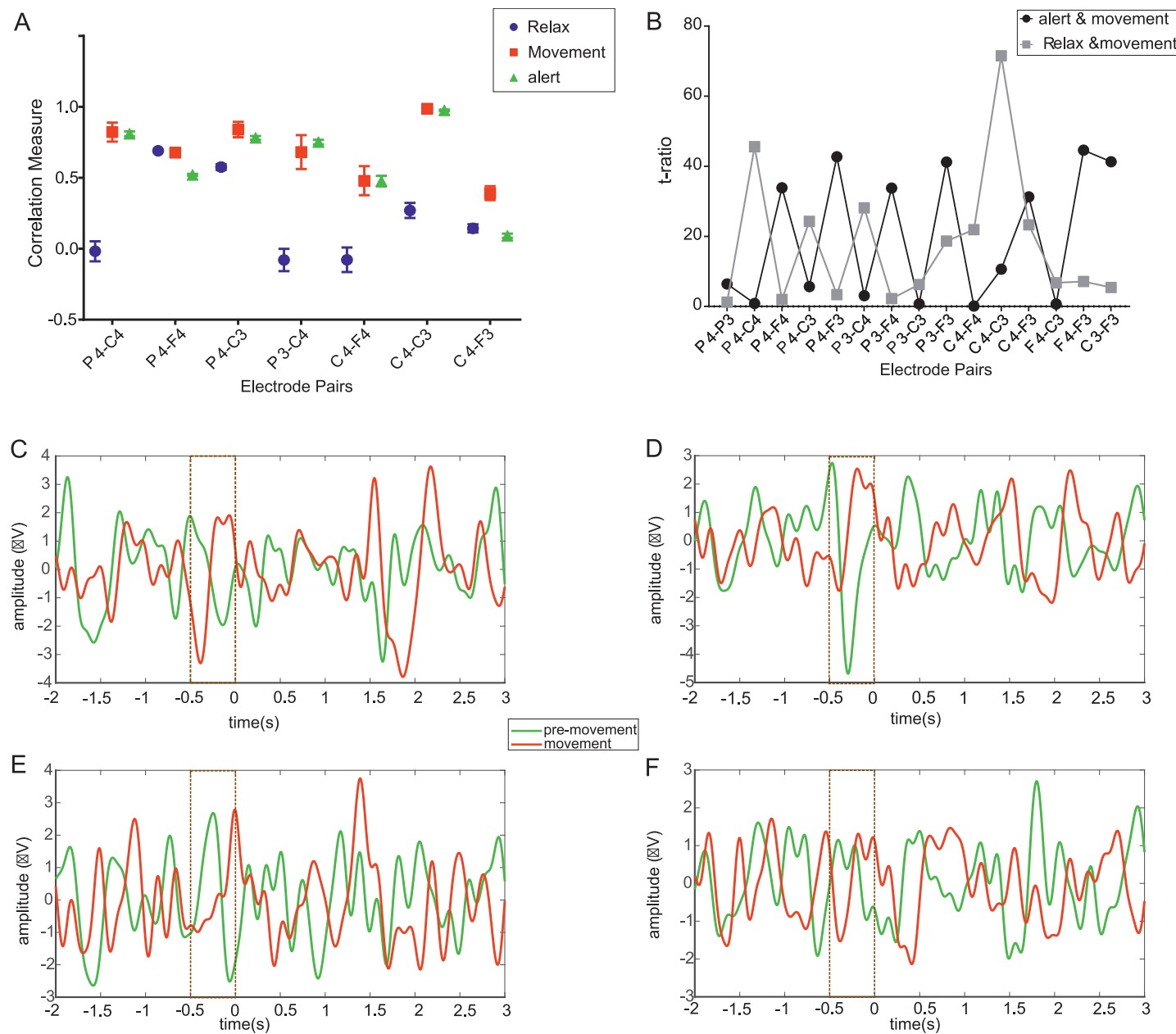

**Fig 3. Temporal correlations across MRCPs from EEG electrode regions differentiate movement tasks.** (A) Correlation measures for central regions (C3-C4) electrode pair has indicated higher correlation (r = 1) for movement and 'alert' condition tasks whereas for a lower correlation measure (r) of 0.3 was estimated for the 'relax' task., P4-F4 electrode pair has indicated a measure of 0.5 for premovement and relax tasks and 0.6 for movement task. The electrode combination of C3-C4 for movement and "alert" condition had higher correlation measure compared to P4-C4, P4-F4, P4-C3, C4-F4, C4-F3 combinations. (Blue circle–"relax task", Red square–"movement task", Green triangle–"alert/premovement" task (B)Correlation measure of various electrode pair combinations was statistically compared for the tasks using t-test. and C3-C4 electrode combination were significantly different for "relax" and movement tasks indicating central region electrodes could be decisive in differentiating "relax" and grasped movement tasks. (C) potential amplitude decreases in 296% from premovement to movement in the central electrode regions (D) Parietal regions has shown 164% decreased in amplitude (E) Frontal regions has shown 133% decrease and (F) Occipital regions has shown 230% decrease in potential amplitude before 500ms to onset of movement. (Green line–"premovement task" and Red Line indicates "movement" task).

A t-statistic value of 70 for the "relax" and the movement tasks at the central electrode regions indicated that the two tasks were significantly different from each other for C3-C4 (Fig 3B). P4-C4 combination of electrode regions reported a t-statistic of 45 for "relax" and "movement" tasks and P3-F4 and P4-F4 regions indicated a t-statistic value of less than 10. F4-C3, F4-F3, and C3-F3-central regions showed t-statistic values of less than 10 suggesting the tasks may not be different from each other across these electrode pairs. Among the central (Fig 3C), parietal (Fig 3D), frontal (Fig 3E), and occipital regions (Fig 3F), although no change in amplitude of the MRCP was observed during "alert" or "premovement" condition, a shift from the negative to positive wave was observed for movement tasks before movement (-0.5s) until the onset of movement (Fig 3).

## 3.2 Increased motor cortical β and γ frequency bands act as biomarkers for detecting movement during a grasped movement task

Scalp topographies indicated the presence of the attenuated α band (ERD) and amplified β band (ERS) modulations during movement initiation in the central parietal regions over the different sub-band regions (see S1 Fig in S1 File). 15–25 Hz β sub-band regions showed activation along the precentral gyrus, postcentral gyrus (corresponding to Central C3, C4 electrode locations) (See S1 Table in S1 File for more electrode locations), and superior lateral occipital cortex (corresponding to parietal P3, P4 electrode locations) (see S1A and S1B Fig in S1 File).

**3.2.1. Characterisation of μ/α oscillations.** A 26% decrease in the μ/α oscillations was estimated for initiation to grasp movement, a 16% decrease was observed during grasp movement to post-movement and a 29% decrease was prominent during "relax" to grasp movement tasks across central electrode regions (Fig 4A). Compared to the central regions

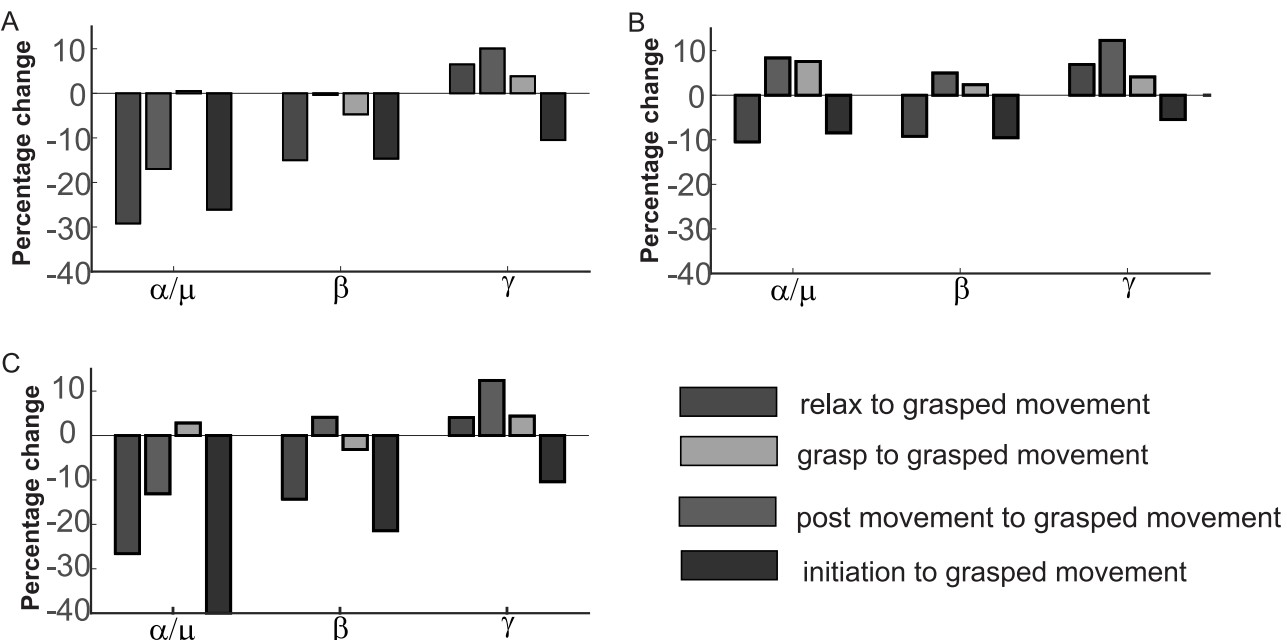

**Fig 4. Increased post movement activity by rebound β and γ oscillations.** (A) α/μ oscillations show 26% decrease from initiation to grasp movement 29% decrease from relax to grasp movement and 16% decrease from grasp movement to post movement in the central electrode regions. 14% and 10% decrease in beta and gamma oscillations from initiation to grasp movement and rebound increment of 14% beta and 20% gamma for post grasp movement. (B) Frontal region has shown 7% increase of alpha oscillations from grasp to grasped movement and 8% increase post movement (C) Parietal region electrodes has shown 21% and 20% decrease from initiation to grasp movement and rebound increment of 4% and 12% post movement. (For inter subject variability see S3 Fig in S1 File).

(corresponding to C3-C4 positions), frontal regions (corresponding to F3-F4 positions) reported a 7% increase in μ/α oscillations for grasp to grasped movement, and an 8% increase post-movement (Fig 4B).

**3.2.2. Characterisation of β and γ.** There was a decrease of 14% of β and 10% of γ oscillations during movement initiation to grasped movement, and a rebound increment of 14% of β and 20% of γ frequencies post-movement were observed (Fig 4A). 9% decrease of β and 5% decrease of γ rhythms from initiation to grasped movement and rebound increment 4% and 12% for β and γ oscillations were observed in the frontal regions. In the parietal regions, 21% for β and 10% for γ decreased and rebound increment of 4% and 12% for β and γ oscillations. (Fig 4C). Also, a 2% increase for grasp to grasped movement and a 13% decrease post-movement for μ rhythms (Fig 4C) were observed.

A 7% increase in θ oscillations was observed post-movement in the middle frontal gyrus and frontal pole (corresponding to frontal lobes). During the task change from grasp to grasped movement, theta oscillations decreased by 20% in the central region (precentral gyrus and postcentral gyrus) and 30% in the parietal (superior Lateral Occipital Cortex) regions (See S2 Fig in S1 File).

The three tasks with the oscillations (α, β, and γ) were compared using the $\chi^2$- test. The $\chi^2$ value = 21.51 (degree of freedom = 4), indicated the percentage of brain oscillations varied among premovement, left movement, and right movement tasks rejecting the presumed null hypothesis and were statistically significant (p = 0.003). Among genders, with $\chi^2$ statistic as 89.43 (degree of freedom = 8), p <0.0001, the data showed significant variations among male and female subjects (see S2 Table in S1 File).

The Tukey test (see S3 Table in S1 File) among male and female subjects indicated that β oscillations of right-hand grasped movement from male participants were significantly different from β oscillations of premovement, left hand grasped movement, and right-hand grasped movement from female participants (p-value <0.0001) (See Fig 5).

## 3.3 Machine learning and task-specific classification of grasped movement

Classification for discriminating tasks on the EEG datasets from the 14-channel electrode and 32 channel electrodes across 170 samples and model evaluation was performed. SVM (polynomial order 3) gave the best accuracy in most classification tasks, with the best mean accuracy of 60% compared to SVM (with radial basis function), multi-layer perceptron (MLP), and decision tree (DT) algorithms (See S4A Fig in S1 File). With the test dataset of 100 samples, the SVM polynomial kernel had a higher performance rate of 54% compared to other algorithms (see S4B Fig in S1 File). When comparing left movement and right movement, only 49% training accuracy was noted with SVM (see S5 Table in S1 File for accuracy rate with other algorithms).

The premovement and movement tasks dataset acquired from central regions (50 training samples) allowed to generate a statistically significant model of classification with 70% training accuracy (see Fig 6A) and 50% (6 samples) test accuracy (see Fig 6B) using SVM and MLP algorithms. In the case of the left- and right-hand movement 76% training accuracy and 83% (F1 = 0.83; AUC = 1) test accuracy with the SVM polynomial kernel. (See S6 and S7 Tables in S1 File for accuracy, AUC and F1 scores) was observed. For the 20-features dataset generated with feature ranking methods, a 16% error rate was obtained when using decision trees and by pruning the ranked 20 features to 5 leaf nodes, and with minimum samples to 3 (see S5 Fig in S1 File for accuracy with 20 ranked features dataset).

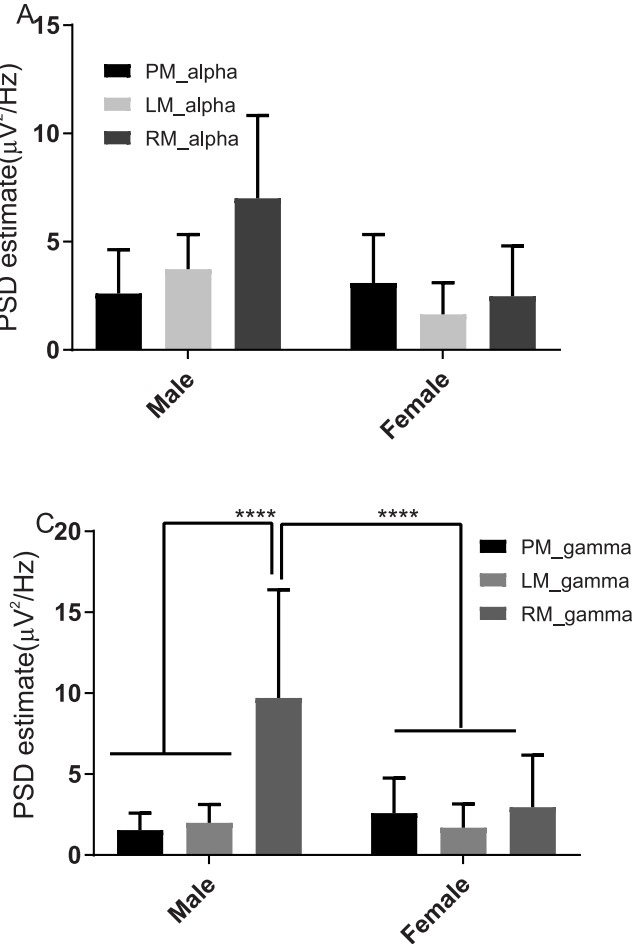

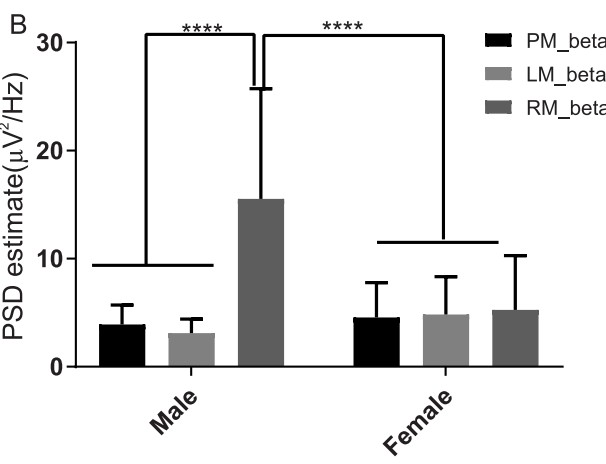

**Fig 5. Significant left and right movement in male and female subjects in the central region electrodes among β & γ oscillations.** (A, B, C) Variability between the male and female subjects for the tasks of premovement (PM) left movement (LM) and right movement (RM). (A) α oscillations for premovement task of male subjects (2.609 ± 2.025) female subjects (3.091 ± 2.245) have not shown any statistical significance similarly for left movement and right movement (p value >0.9999), (B) β and (C) γ oscillations have shown significant differences were presented with p value < 0.0001 (12 sample trials of Male subjects, 12 sample trials of female subjects).

## 4. Discussion

In this study, an exploration of grasped arm movement dissecting the pre-movement and movement-specific MRCP and ERD/ERS morphology was performed in the context of multiple movement tasks. Interpreting EEG underlying left or right arm movement, allowed learning models to discriminate movement intention or execution for clinically relevant assessments and can be used for current and future brain-machine interfaces. Specifically, in this study, feature-based interpretations of grasped movement and premovement have been highlighted within EEG data using the underlying low-frequency time-domain data characteristics & from the γ oscillations observed in the central cortical regions.

The combination of the temporal peak of the readiness potential peak and the μ, β, and γ oscillations can be used to precisely represent premovement and grasped movement tasks. The signal's temporal shift in deflection towards the positive before the onset of movement can be used as a potential biomarker to differentiate movement intention and grasped movement. Central regions' importance in controlling arm movements may be justified as the accuracy reported among those electrodes was higher.

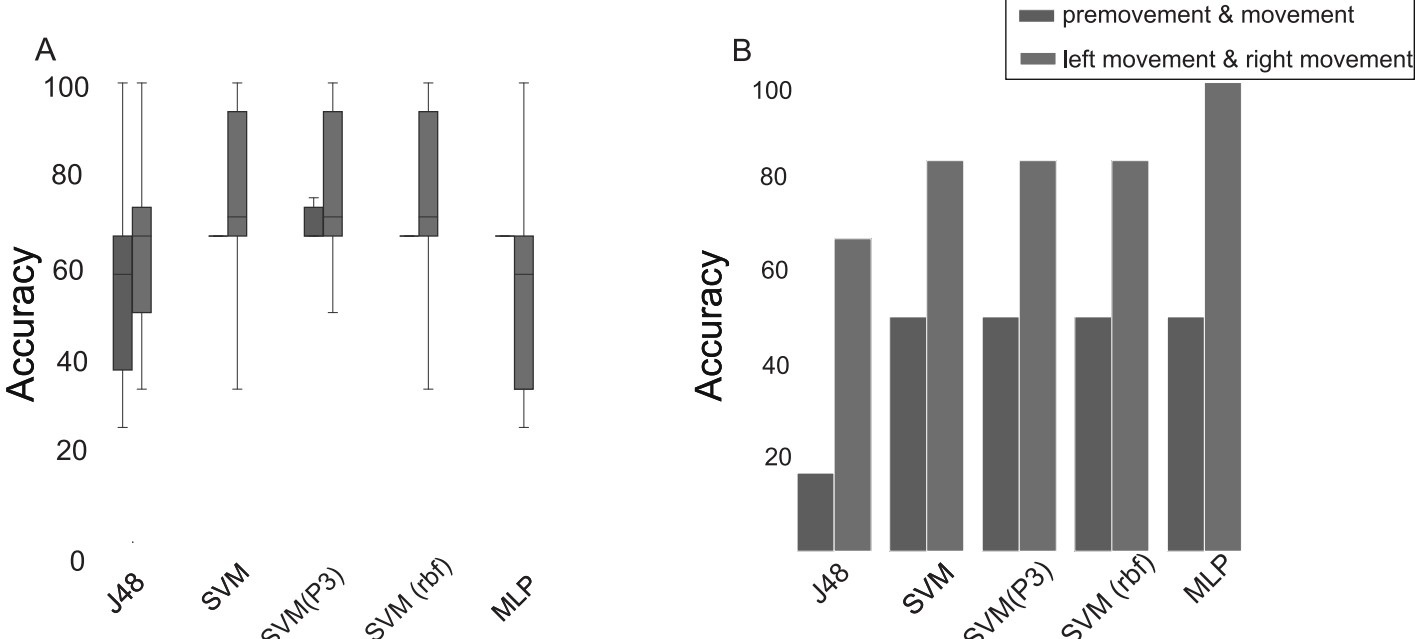

**Fig 6. Machine learning-based model performance.** (A) Training accuracies for central regions on using SVM had outperformed MLP and DT for left and right hand movement (SVM: Support Vector machine; SVM-(P3): support vector machine polynomial degree 3, SVM (rbf): Support vector machine—radial basis function, MLP: Multilayer Perceptron) (B) Testing accuracy performance with central regions SVMs has performed similarly with 50% accuracy for premovement & movement, and 80% accuracy for left and right hand movement classification DT has performed moderately for left vs right hand movement.

The main differences between premovement and movement may be observed within the first 0.5 s before the movement onset, mainly over the contralateral primary motor cortex (locations C3, C4). Our results suggest that the attenuated α oscillations and increased β oscillation topographies at central and parietal regions were indicative of hand movement states. A decrease in α, β, and γ-band activity compared to the "relax" (resting-state) task could be indicative of the progression of the grasped movement, and a rebound β and γ frequency activity after grasped movement can serve as spectral biomarkers of pre and post-movement assessments. In the C3 and C4 electrode regions, other than the μ and β bands, the γ band was also associated with grasped movement. Pre-movement and grasped movement conditions were significantly different in the central regions implying that the central regions may provide optimal features to train and predict BCI-related classification models. Although some of the frontal electrodes were evaluated, better accuracy observed with C3-C4 electrodes in the central regions corresponds to the capacity of discrimination of left- and right-hand movement by these overlapping hand regions. The β and γ-band variations across asymmetry may be crucial for discriminating left- and right-hand grasped movement and the pre-movement suppression and post-grasped movement rebound of β band in the context of the laterality of neural activity especially in the ipsilateral motor cortical areas, which could be relevant for the task-based evaluation of healthy and Parkinson's disease patients.

Through machine learning analysis, model-based discriminators were generated focusing on premovement and grasped movement task classification. Machine learning classifiers could discriminate left and right direction movement using the same features as in premovement and movement tasks. Improved classification accuracy while discriminating pre-movement and grasped movement, was obtained using peak amplitude of readiness potential and the PSD estimates of μ, β, and γ oscillations. In this study, SVM with a polynomial kernel with a

lower order provided the highest accuracy over other machine learning classifiers, and allowed faster model building and testing while discriminating peri-movement and movement data. MLP also performed relatively well, although not as optimally as the SVM compared to several other interpretable classifiers on the EEG datasets. In the case of reducing features, combining decision tree-based feature ranking and applying pruning as pre-processing of data could improve classification accuracy. Estimating across several EEG-based tasks and measurement modalities as the pruning could help generate a more generic learning model that may need to be tested on movement imagery and movement execution-based arm movement datasets.

The artifact removal in EEG involved the removal of eye-blink, eye movements, and tongue movements helped augment the classifier's accuracy and was crucial to remove those data points that had EEG artifacts before evaluating with classifiers. The study also suggests that employing low-cost consumer-grade EEG devices, given their ease of integration and instead of and alongside clinical-grade devices, could capture critical information related to grasped movement and its execution had similar discrimination models and may be valuable in finding candidates for clinical trials.

## 5. Conclusion

Premovement and grasped movement may vary across spatio-temporal scales but discrimination of left and right grasped movement could be performed with temporal and spectral analysis and combining classification methods for decoding pre-movement neural activity in the case of stereotyped left or right arm movements. Interpretation relied on low-frequency time-domain signals and γ oscillations for decoding the left and right movement tasks and with minor variations across genders. As observed in neurological conditions, activations in ipsilateral sensorimotor areas can be used to interpret compensation mechanisms related to the movement implying bi-electrode C3-C4 regions may be task discriminative. Although not presented in this study, the temporal and spectral features can be the foundation for novel control strategies targeting the directional choices of an operational robotic arm.

## Supporting information

**S1 File.**
(PDF)

## Acknowledgments

This work derives direction and ideas from the Chancellor of Amrita Vishwa Vidyapeetham, Sri Mata Amritanandamayi Devi.

## Author Contributions

**Conceptualization:** Sandeep Bodda, Shyam Diwakar.

**Data curation:** Sandeep Bodda, Shyam Diwakar.

**Formal analysis:** Sandeep Bodda, Shyam Diwakar.

**Funding acquisition:** Shyam Diwakar.

**Investigation:** Sandeep Bodda, Shyam Diwakar.

**Methodology:** Sandeep Bodda, Shyam Diwakar.

**Project administration:** Shyam Diwakar.

**Resources:** Shyam Diwakar.

**Software:** Sandeep Bodda.

**Supervision:** Shyam Diwakar.

**Validation:** Sandeep Bodda, Shyam Diwakar.

**Visualization:** Sandeep Bodda, Shyam Diwakar.

**Writing – original draft:** Sandeep Bodda, Shyam Diwakar.

**Writing – review & editing:** Sandeep Bodda, Shyam Diwakar.

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
