## [Decision Letter · Decision Letter 0]

14 Mar 2022

PONE-D-21-37461Exploring Temporal Dynamics and Spectral Oscillations including Gamma Rhythms in EEG Resolutions underlying Grasped Hand MovementPLOS ONE

Dear Dr. Diwakar,

Thank you for submitting your manuscript to PLOS ONE. After careful consideration, we feel that it has merit but does not fully meet PLOS ONE’s publication criteria as it currently stands. Therefore, we invite you to submit a revised version of the manuscript that addresses the points raised during the review process.

The revised work needs to satisfy the concerns raised by the reviewers.

Please submit your revised manuscript by April 20, 2022. If you will need more time than this to complete your revisions, please reply to this message or contact the journal office at plosone@plos.org. Please include the following items when submitting your revised manuscript:A rebuttal letter that responds to each point raised by the academic editor and reviewer(s). You should upload this letter as a separate file labeled 'Response to Reviewers'.A marked-up copy of your manuscript that highlights changes made to the original version. You should upload this as a separate file labeled 'Revised Manuscript with Track Changes'.An unmarked version of your revised paper without tracked changes. You should upload this as a separate file labeled 'Manuscript'.

We look forward to receiving your revised manuscript.

Kind regards,

Mukesh Dhamala, Ph. D.

Academic Editor

PLOS ONE

Journal Requirements:

2. Please provide additional details regarding participant consent. In the ethics statement in the Methods and online submission information, please ensure that you have specified what type you obtained (for instance, written or verbal, and if verbal, how it was documented and witnessed). If your study included minors, state whether you obtained consent from parents or guardians. If the need for consent was waived by the ethics committee, please include this information

Reviewers' comments:

Reviewer's Responses to Questions

**Comments to the Author**

1. Is the manuscript technically sound, and do the data support the conclusions?

Reviewer #1: Yes

Reviewer #2: Yes

2. Has the statistical analysis been performed appropriately and rigorously? 

Reviewer #1: No

Reviewer #2: No

3. Have the authors made all data underlying the findings in their manuscript fully available?

Reviewer #1: No

Reviewer #2: No

4. Is the manuscript presented in an intelligible fashion and written in standard English?

Reviewer #1: No

Reviewer #2: No

5. Review Comments to the Author

Reviewer #1: 1- The title is too long, and can be truncated to: Exploring EEG spectral and temporal dynamics underlying a hand grasp movement

2- The abstract should be revised for grammatical, structural and typing errors, such as:

The data acquisition was done for 163 trails of 30 ...

Instead, say: The EEG data was acquired from 30 participants, where each performed 163 trials of a hand grasping movement.

3- The introduction should be grammatically re-structured, too many faults.

ex: Neural correlates to voluntary grasped movement .. should be: Neural correlates of

4- In the methods: mean age +/- std ?

Amrita Vishwa Vidyapeetham: If this is the name of the university, please write 'university'.

In this study, a 32-electrode and a 14-electrode recording systems were employed: which one exactly ? and why using 2 systems instead of either one ?

total of 173 trials were recorded for each task: for each participant ?

Every subject participated in two sessions of 4 trials and every trial was separated by two minutes of resting: 4 trials or 173 ?

what is is total duration of the experiment ?

what is the total duration of each trial ?

in line 174: the ”, for what ?

The grasp should be done in how many seconds ? how the participant knew the start of a new trial ? hwo does the system knew the end of a trial and the beginning of a new trial ?

2.3. Data Acquisition: here you say 32 electrodes!

line 188: is it the raw EEG data ?

line 224: what is the length of each sample ? what are these samples (features) ? only filtered EEG data ? spectral ? in which EEG bands ?

PLEASE re-write the methods section as many details are not present.

PLEASE re-write the discussion section for structural errors

Reviewer #2: This paper presents the analysis and classification of EEG signal recorded during grasp movement. The work consisted of an experimental task were healthy participant moved any of the arms in a synchronized experiment. One critical problem with this study is the lack of novelty, many aspects are presented in previous literature. Other problem is with analyses which are very poor and require significant improvement.

Some comments

1) What is the motivation to carry out this research? What is the scientific problem you are addressing?

2) Methods section: 1st paragraph: were the 173 trials recorded for each participant or for all of them? What are the tasks?

3) Experimental Design & Procedure: as I understand, each subject participant in two session and four trials were recorded per session.

4) In general, the description of the experimental task. For instance, “Experimental Design & Procedure:” did not presented anything about the use of both hand and two directions, but this information is subsequently presented in “Trial structure:”. Authors should improve the description of the experiment. In consequence, it seem that only two trials for hand and direction are recorded.

5) In the 1st paragraph of “Methods” you said that “In this study, a 32-electrode and a 14-electrode recording systems were employed.”, but then in “2.3. Data Acquisition:” you said that “we used a 32-electrode commercially available device”. This is highly confusing.

6) In many places there are not space between words and references, while in other places there are. Or the reference is after the end point of a sentence. Please review.

7) “The relative band power was estimated based on the brain activity bands for left hand movement and right-hand movement tasks.”: this seems to be a critical step that required further explanations. Its is not possible to replicate the procedure in its current state

8) More details and/or results about the use of ICA to remove artefactual EMG components embedded in the EEG should be given.

9) The data analysis descriptions is very plain and this is an important aspect of the research. In its current state is it not possible to replicate the procedures therein.

10) In the preprocessing of the data, why did you apply two filtering steps? I do not see the need for that since at the end only the BPF has an impact on the signals

11) The methods do not show how the MRCP are extracted from the preprocessing EEG signals.

12) It is not clear what you did in “2.5. Statistical measures:”. This needs significant rewording. In consequence, it is nor clear how the statistical analysis were done.

13) There are organization problems. One the is that you call figure 1, then figure 5.

14) “The dataset contained 270 samples has a sampling rate of 128 Hz and 56 samples 225 dataset has a sampling rate of 500 Hz.” Why are there two sampling frequencies?

15) I am not sure there you have MRCP since you are removing all frequency content below 1Hz.

16) What time windows of the trial were used to classify? What are the number of classes/categories? What is the reason to employ several classifiers? What is the dimension of the input vector? There are several open questions in the machine learning analysis.

17) Why to decode movement since in real BCI setting with final users (patients) they are not able to move or at best they only have residual movements (obviously depending on the medical condition)

18) The number of trials is very limited for each participant

19) It seems you are combining data from all participants. This is not a common strategy in the analysis of EEG signals, and they are usually done subject specific.

6. PLOS authors have the option to publish the peer review history of their article (what does this mean?). If published, this will include your full peer review and any attached files.

Reviewer #1: **Yes: **Bilal Alchalabi

Reviewer #2: No

---

## [Author Response · Author response to Decision Letter 0]

27 May 2022

We thank the editor and the reviewers for their constructive and valuable comments, and we have used the same to address changes in the manuscript. We have included additions also to our supplementary material. 

Reviewer 1: 

1. The title is too long, and can be truncated to: Exploring EEG spectral and temporal dynamics underlying a hand grasp movement

The title has been modified to "Exploring EEG spectral and temporal dynamics underlying a hand grasp movement", as suggested by the reviewer. Thank you for the suggestion. 

2. The abstract should be revised for grammatical, structural, and typing errors, such as:

The data acquisition was done for 163 trails of 30. 

Instead, say: The EEG data was acquired from 30 participants, where each performed 163 trials of a hand grasping movement.

Thank you for the comment. The abstract has been revised, and the manuscript was checked for grammatical and language errors and typos. The abstract has been slightly modified (line number 16 to 38 on page number 2) to reflect these adaptations. 

3. The introduction should be grammatically re-structured, too many faults.

ex: Neural correlates to voluntary grasped movement. should be: Neural correlates of

Thank you. Agreeing with the reviewer, we have revised the manuscript. 

As suggested by the reviewer, the statement on page number 4, line 49 have been rewritten as "Neural correlates of voluntary grasped movement are relevant in the development of modern prosthetics, and towards Brain Computer Interface (BCI) research (Daly and Wolpaw, 2008; Lebedev and Nicolelis, 2006; Wolpaw et al., 2000)." 

4. In the methods: mean age +/- std?

Thirty healthy volunteers aged 18 to 30 years (mean age = 22.32 ± 1.92 years) participated in this study. This has been added to the methods section on page 9, line number 177) 

5. Amrita Vishwa Vidyapeetham: If this is the name of the university, please write 'university'.

As per Indian legal records, the university is called "Amrita Vishwa Vidyapeetham", where "Vishwa Vidyapeetham" is the Sanskrit term for university. However, for practical reasons, as indicated by the reviewer we have used the term "University". The corrections can be seen on page number 9 at line number 179 in the manuscript. 

6. In this study, 32-electrode and 14-electrode recording systems were employed: which one exactly? and why using 2 systems instead of either one? 

In this study both 32 -electrode medical graded system and 14 electrode consumer graded systems have been used. The consumer graded 14 electrode device is equipped with Frontal, temporal, parietal and Occipital sensors (AF3,AF4, F3,F4,F7,F8,FC5,FC6,P7,P8,O1,O2,T7,T8) and the medically graded 32 electrode device is equipped with 32 sensors from the same regions including central region (FP1,FP2,AF3,AF4,F3,F4,Fz,FC1,FC2,F7,F8,FC5,FC6,C3,C4,Cz,CP1,CP2,CP5,CP6,T7,T8,P7,P8,P3,P4,PO3,PO4,Pz,O1,O2,Oz ). Based on the study of comparison of medical and consumer EEG systems by Ratti et al., 2017 consumer systems were prone to artifacts in frontal regions, medical devices offer clear advantages in data quality, reliability, and depth analysis over consumer systems.

Since the study is more focused in clinically aspects and application of neuro prosthesis at consumer level, it was crucial to look at the accuracy from obtained features using both systems and to map neural activity that were not restricted to frontal and temporal regions only.

Though 32-electrode can provide a greater dataset compared to the 14-electrode, the 32-electrode device was more expensive (computationally). As researchers based in a developing country like India, we also are focused on using and implementing how low-cost devices can be used reliably. With this as a possible alternative for recordings outside lab settings, we have checked whether 14-electrode device provided similar or reliable accuracy like that of a 32-electrode device. The results suggest that features from central electrode regions provide reliable accuracy for premovement and movement tasks, compared to frontal electrode regions. The clear discrimination of MRCP pattern for movement initiation was observed from the central region electrodes. 

In the present manuscript, introduction section on page 7, line number 135 to 140 and on page 8 line 156 to 166 we have now modified the text with description. We thank the reviewer for the comment. 

7. total of 173 trials were recorded for each task: for each participant? Every subject participated in two sessions of 4 trials and every trial was separated by two minutes of resting: 4 trials or 173?

In this study, we have recorded 175 trials from 30 subjects. Off these 30 subjects, 10 subjects were recorded using 32-electrode system. For each subject, four trials per task were carried out. 40 trials were performed with 10 subjects. Due to the loss of data packets, EEG data of 12 trials were discarded. For 20 subjects, the 14-electrode system was used. With that 14-electrode device, 10 trials were performed for 6 subjects, 5 trials for 12 subjects, 13 trials for 1 subject, and 2 trials for 1 subject have been carried out and EEG signals were recorded. The recording for each task was taken in two separate sessions, for LHLD, RHRD tasks

We have included this information in the present draft on page 11, line number 233. We thank the reviewer for the comment. 

8. what is total duration of the experiment? what is the total duration of each trial

Each recording employed in this study was for 45 seconds and considered as a trial of the experiment. We have reedited the methods in the manuscript (line number188, page number 10) as read below. 

A cue-based paradigm (Fig 1B) was employed where subjects were presented with visual cues using a slide presentation of the duration of 45 seconds. we have recorded each trial as an experiment. EEG signals were recorded for 10 subjects using a 32- electrode device. For each subject, four trials were performed per task. So, a total of 40 trials were performed for 10 subjects. Due to the loss of data packets, EEG data of 12 trials were discarded. Using the 14-electrode device, EEG signals were recorded for 20 subjects. In this, 10 trials were performed for 6 subjects, 5 trials for 12 subjects, 13 trials for 1 subject, and 2 trials for 1 subject have been carried out and EEG signals were recorded. Out of 175 trials, 163 were used for the data analysis, and 12 trials were rejected due to the loss of signal packets during the recording.

9. in line 174: the", for what?

Thank you for pointing out a typo, we have edited the manuscript (page number 10, line number 205-220)

10. The grasp should be done in how many seconds? how the participant knew the start of a new trial? How does the system knew the end of a trial and the beginning of a new trial?

The protocol has been recorded for 45 seconds duration with different cues, The Relax cue was show for 10 seconds, alert cue was shown for 5 seconds, reach cue was shown for 5 seconds, grasp cue was shown for 5 seconds, movement cues were shown for 5 seconds and stop cue with blank screen shown for 5 seconds. The next trial was started after 3 minutes break. The blank screen cue presentation indicates the beginning and end of the trials. 

This has been incorporated in the Methods sections, Experimental Paradigm on page 10, Line numbers 205-220 clarified with different steps of the protocol in the manuscript. 

11. Data Acquisition: here you say 32 electrodes!

We have now edited the methods to clarify this point as well. 

As indicated previously, in this study, we have recorded from two different set of experimental devices with frontal and central electrode sensors and have analyzed data separately. The results suggest that features from central electrode regions have more accuracy prediction rate for premovement and movement tasks, compared to frontal electrode regions. The clear discrimination of MRCP pattern for movement initiation was observed from the central region electrodes. 

For clinically relevant data, we used a 32-electrode commercially available device (Neuroelectrics, Barcelona, Spain) positioned on the scalp according to the 10–20 international system with a sampling rate of 500Hz. Considering commercial and limited electrode platforms, we also employed a 14-electrode device (EMOTIV EPOC+) with a sampling rate of 128 Hz. EEG signals were recorded for 10 subjects using a 32- electrode device. For each subject, four trials were performed per task. So, a total of 40 trials were performed for 10 subjects. Due to the loss of data packets, EEG data of 12 trials were discarded. Using the 14-electrode device, EEG signals were recorded for 20 subjects. In this, 10 trials were performed for 6 subjects, 5 trials for 12 subjects, 13 trials for 1 subject, and 2 trials for 1 subject have been carried out and EEG signals were recorded. Out of 175 trials, 163 were used for the data analysis, and 12 trials were rejected due to the loss of signal packets during the recording. 

Thank you for the question. 

12. line 188: is it the raw EEG data?

We refer to the raw data acquired from the device was detrended to minimize the drifts and further band pass filtered 1- 60 Hz range. This now reads edited as line 246 on page 12. Here is the edited text from the manuscript. 

To obtain the spectral components initially the data was high pass filtered (1Hz) to minimise the drifts [88]and the reference (mean) was subtracted, further the data was detrended [89] and bandpass filtered using an FIR filter [90,91] of order 20 within the range, 1Hz - 60 Hz and notch filter was applied to remove line noise in the range of 50 Hz. 

13. line 224: what is the length of each sample? what are these samples (features)? only filtered EEG data? spectral? in which EEG bands?

We have used two different datasets from two different devices, the first dataset with 56 samples (no. of rows/ no. of trials) with a binary class of premovement & movement consists of 74 columns which are features (readiness potential peak (component of MRCP), alpha band, beta band, and gamma-band). The second dataset with 270 samples (no. of rows / no. of trials) consists of 67 columns which are features.

The features are readiness potential peak and spectral oscillations of alpha, beta and gamma band obtained after preprocessing the raw EEG data. 

14. PLEASE re-write the methods section as many details are not present. 

Thank you for the comment. We have rewritten the methods section as suggested. 

We have edited the “Experimental Design & Procedure” completely restructuring it as Experimental Paradigm. The present draft in the manuscript from the line number 187 to 230 has been re-addressed with stepwise procedure to perform the experiment and has included number of subjects considered for data acquisition and number of trails recorded in total from all the subjects. We have also modified the trial structure part as a step-by-step method for ease of clarity and understanding (from line number185 to 225 at page number 11)

In the methods section the offline processing data was added and indicates our methods for power spectral analysis and movement related cortical potentials. (Page number 12, Line number 244-292). 

The statistical analysis was also re-structured and elaborated (page number 14 line numbers 293- 332 in the manuscript). 

15. PLEASE re-write the discussion section for structural errors

We apologize for the issues in the previous draft and thank the reviewer for the comment. We have avoided structural issues (page 21 at line numbers 457 -513) and redrafted the discussion section.

Reviewer 2:

1. What is the motivation to carry out this research? What is the scientific problem you are addressing?

We thank the reviewer for this question. Our motivation for this study is now part of the introduction in the present manuscript on page 4, line number 43 -48 and on page 8, line number 159-166

Patients with amyotrophic lateral sclerosis (ALS) or spinal cord injury (SCI) are reported to have significant loss of voluntary motor control and extensive dysfunction of upper and lower limbs [1–4]. Rehabilitation using robotic or functional electrical stimulation could facilitate novel therapeutic strategies for such conditions. Integration of signals could also be a potential goal for Brain-Computer Interfaces (BCI), allowing the transfer of control information from a prosthesis onto the brain tissue [5–7]. Neural correlates of voluntary grasped movement are relevant in the development of modern prosthetics and towards Brain-Computer Interface (BCI) research [8–10]. Insights from electroencephalography (EEG) recordings and their underlying neurophysiological processes involved in motor tasks can help decode movement and its functional interpretation. EEG signal components related to movement allow noninvasive measurements, making them suitable for natural BMIs [5,6]. In addition, these correlates are measurable in paralyzed patients [3,4,7,11]. In this study, we explored the EEG dynamics by employing a novel protocol to understand movement intention and execution during grasped movement tasks performed in the left and right directions. Exploring the activity space underlying grasping controlled by parieto-frontal circuits [86] and central regions, this paper also attempts to relate the variations in movement initiation and grasped movement by developing classification models based on features of temporal and spectral correlations specifically with a combination of α, β, and lower � frequency bands and readiness potentials for grasped-movement. 

2. Methods section: 1st paragraph: were the 173 trials recorded for each participant or for all of them? What are the tasks

We thank the reviewer for the question. We have redrafted methods to reflect the trials, questions, and procedure. 

In this study, 175 trials recorded for all the participants., The tasks included left hand grasped left direction movement (LHLD), a right hand grasped right direction movement (RHRD), and a premovement. These tasks followed timelines based on the cueing paradigm and were performed by subjects for 45 seconds for each trial.

As per the reviewer’s suggestions, we have reedited the methods and have included step by step experimental paradigm in the manuscript on page 10, line number 187-225. 

The experimental paradigm was conducted as outlined in the following steps. 

1. All trials commenced with a relaxation phase (blank screen), considered as a reference or baseline signal for the analysis

2. The subject was asked to relax for ten seconds.

3. The subject was alerted for the following task by a ‘+’ sign cue for 5 seconds. 

4. The word ‘Reach’ as a cue was shown for 5 seconds, indicating the subject to reach the object. 

5. Then an image of a bottle was shown to the subject, indicating to grasp the bottle placed in front of the table. This cue was presented for five seconds 

6. An upward arrow cue (↑) was shown for next 5 seconds indicating to the subject to lift the bottle to the chin level. 

7. This was followed by a leftwards arrow cue (←) for five seconds; subject was then instructed to move the bottle towards left direction using the left hand.

8. Following this, a down-wards arrow cue (↓) was shown for 5 seconds indicating to the subject to place the object back down on the table

9. A blank screen was shown as an indication of the end of the experiment trial

The same steps were repeated for right-hand right direction movement task as well. 

The tasks were carried out using both hands for the two movement directions in trials defined as right-hand leftward direction (RHLD), right-hand rightward direction (RHRD), and left-hand leftward direction (LHLD), and left-hand rightward direction (LHRD). However, only LHLD and RHRD have been considered for the analysis.

The same steps were repeated for right hand right direction of movement 

3. Experimental Design & Procedure: as I understand, each subject participant in two sessions and four trials were recorded per session.

In this study, 175 trials recorded from 30 subjects. Out of these 30 subjects, 10 subjects were using 32-electrode system. For each subject, four trials were performed per task. So, a total of 40 trials were performed for 10 subjects. Due to the loss of data packets, EEG data of 12 trials were discarded. Yes, each participant has participated in two sessions to acquire the two separate directions of movement left-hand left direction movement and right-hand right direction movement, the recording for each task was taken in two separate sessions, (LHLD, RHRD). This has been incorporated in the manuscript methods section on page number 11 from line number 233 can be read as below paragraph.

EEG signals were recorded for 10 subjects using a 32- electrode device. For each subject, four trials were performed per task. So, a total of 40 trials were performed for 10 subjects. Due to the loss of data packets, EEG data of 12 trials were discarded. Using the 14-electrode device, EEG signals were recorded for 20 subjects. In this, 10 trials were performed for 6 subjects, 5 trials for 12 subjects, 13 trials for 1 subject, and 2 trials for 1 subject have been carried out and EEG signals were recorded. Out of 175 trials, 163 were used for the data analysis, and 12 trials were rejected due to the loss of signal packets during the recording

4. In general, the description of the experimental task. For instance, "Experimental Design & Procedure:" did not present anything about the use of both hands and two directions, but this information is subsequently presented in "Trial structure:". The authors should improve the description of the experiment. In consequence, it seems that only two trials for hand and direction are recorded.

Thank you for the suggestions, we have restructured the experimental design & procedure into a single structure called experimental paradigm and have included in the manuscript on page number 10-11 lines 187 – 225.

5. In the 1st paragraph of "Methods" you said that "In this study, a 32-electrode and a 14-electrode recording systems were employed.", but then in "2.3. Data Acquisition:" you said that "we used a 32-electrode commercially available device". This is highly confusing.

This was an important choice for us and we thank the reviewer for raising the question. Both 32 -electrode medical graded system and 14 electrode consumer graded systems have been used in this study. The consumer grade 14 electrode device was equipped with Frontal, temporal, parietal and Occipital sensors (AF3,AF4, F3,F4,F7,F8,FC5,FC6,P7,P8,O1,O2,T7,T8) and the medical grade 32 electrode device was equipped with 32 sensors from the same regions including central region (FP1,FP2,AF3,AF4,F3,F4,Fz,FC1,FC2,F7,F8,FC5,FC6,C3,C4,Cz,CP1,CP2,CP5,CP6,T7,T8,P7,P8,P3,P4,PO3,PO4,Pz,O1,O2,Oz ). Based on the study of comparison of medical and consumer EEG systems by Ratti et al., 2017 consumer systems were prone to artifacts in frontal regions, medical devices offer clear advantages in data quality, reliability, and depth analysis over consumer systems.

Since the study was focused on interpreting signal aspects for evaluating pre and movement information at consumer level, it was helpful to look at the accuracy of obtained features using both systems and in order to map neural activity that was not related to frontal and temporal regions.

The 32-electrode could yield more data compared to the 14-electrode and yet, the 32-electrode device was more expensive (computationally and financially). Apart from this paper, it was crucial to evaluate the applicability of low-cost devices especially in off laboratory settings. With this as an additional reason, we employed the 14-electrode device and tested if results were reliable as performed on a 32-electrode. Our results suggest that features from central electrode regions yielded better accuracy for premovement and movement tasks, compared to frontal electrode regions. The discrimination of MRCP patterns for movement initiation was observed from the central region electrodes. 

In the present manuscript, introduction section on page 7, line number 135 to 140 and on page 8 line 156 to 166 we have now modified the text with description. 

6. In many places there are not space between words and references, while in other places there are. Or the reference is after the end point of a sentence. Please review.

Thank you for pointing out this issue. We have corrected the same in the present draft of the manuscript (line numbers 59, 82,83, 94,131, 248,250, 334,335,339).

7. "The relative band power was estimated based on the brain activity bands for left hand movement and right-hand movement tasks.": this seems to be a critical step that required further explanations. It is not possible to replicate the procedure in its current state

We thank the reviewer for pointing out our short sightedness in this regard. Taking the reviewer’s suggestion, the methods section has been modified and details have been added (line number 277 on page 13). 

8. More details and/or results about the use of ICA to remove artefactual EMG components embedded in the EEG should be given.

ICA analysis was performed to remove any artifacts related to EMG, EOG, from the components. AS per the author's knowledge, the authors haven't observed most of the EMG artefactual components in the data, but mostly in the data the first component has seen EOG artifacts which have been removed during the pre-processing stage. In this study Infomax ICA algorithm was used from python mne package to do ICA analysis. The detailed description of ICA method has been edited in the manuscript on page 12 lines 251- 271. 

9. The data analysis descriptions is very plain and this is an important aspect of the research. In its current state is it not possible to replicate the procedures therein.

Thank you for the comment. The manuscript has been modified to reflect changes in the methods section (per say line number 244 on page 12). The present manuscript has a section on data analysis (as listed under Offline processing of EEG data).

10. In the pre-processing of the data, why did you apply two filtering steps? I do not see the need for that since at the end only the BPF has an impact on the signals. 

We thank the reviewer for the comment. We have revised the methods section to address the filtering aspects (page numbers 12 & 13 at line numbers 247 & 286). In our study, we have used high pass filter above 1Hz to minimise the drifts. Spectral oscillations in the frequency range 1Hz to 60 Hz were estimated. To obtain MRCPs, the EEG data was high pass filtered for the 0.1Hz to 5 Hz range. 

11. The methods do not show how the MRCP are extracted from the pre-processing EEG signals.

Thank you for the comment. We have now edited the methods to include on extracting MRCPs. (page 13, line 285). 

12. It is not clear what you did in "2.5. Statistical measures:". This needs significant rewording. In consequence, it is nor clear how the statistical analysis were done.

We agree with this comment. Thank you. In the present draft of the manuscript, we have rewritten the methods section (page 14 line numbers 294 to 329). 

13. There are organization problems. One the is that you call figure 1, then figure 5.

Thank you for pointing out that oversight from our side. We have modified the manuscript. 

14. "The dataset contained 270 samples has a sampling rate of 128 Hz and 56 samples dataset has a sampling rate of 500 Hz." Why are there two sampling frequencies?

Two different datasets have been used for the analysis obtained from medically graded acquisition device which had 500 Hz sampling rate and other one was a consumer grade device at 128Hz sampling rate. The dataset obtained from consumer grade device contains 270 samples (135 premovement, 135 movement) and the second dataset obtained from medical graded device contains 56 samples/trials (28 premovement/ 28 movement) 

15. I am not sure there you have MRCP since you are removing all frequency content below 1Hz.

We thank the reviewer for this comment. The methods section has been modified to show analysis for spectral components and for MRCPs separately (line number 285 on page 13). To obtain the MRCPs, a bandpass filter of 0.1 Hz – 5 Hz was exployed. High pass filter 1Hz was used to obtain the spectral components. 

16. What time windows of the trial were used to classify? What are the number of classes/categories? What is the reason to employ several classifiers? What is the dimension of the input vector? There are several open questions in the machine learning analysis.

We thank the reviewer for the comment. In this paper, time windows of 10s(premovement), and 30s (Left hand grasped movement / right hand grasped movement) of the trial were used in the classification. We chose binary class labels (premovement, movement), (left movement, right movement) and have used two datasets from 2 devices; with the first dataset dimension is 170 * 67 and the second dataset dimension is 56 * 74. 

We have reported this in our re-edited methods section, which has a Machine learning subsection (page number 16, from line number 330). 

17. Why to decode movement since in real BCI setting with final users (patients) they are not able to move or at best they only have residual movements (obviously depending on the medical condition)

Our expectation here was that we targeted the classification for future tools and prostheses. We have started an ongoing study to allow BCI based transfer of control using robotic devices and within such contexts both peri-movement and movement were crucial. Sensory and Cognitive functions of the brain are only minimally affected by amyotrophic lateral sclerosis (ALS) or spinal cord injury (SCI). Rehab with robotic interfaces is also a potential goal for Brain Computer Interfaces (BCI) leading to a possible control or transfer of control signals from prosthesis to actual brain tissue. 

We included some aspects of this into our introduction section (page number 4 from line 43-59). 

18. The number of trials is very limited for each participant

We agree that the number of trials (175) were limited. This was also due to restrictions imposed by COVID. We have however considered all trials together from the subjects as inter subject modelling generalizing the problem using different datasets from various electrode positions for classification with binary class labels. We do hope to extend this study with other BCI components and with our 256-electrode device as we limp back into some form of normalcy after the COVID lockdowns. 

The number of trials and the subjects has been mentioned in the present manuscript at methods section on page 11 from line numbers 227 – 242.

19. It seems you are combining data from all participants. This is not a common strategy in the analysis of EEG signals, and they are usually done subject specific.

Thank you for the comment. For the relevance of commonality across same tasks, we have looked into inter-subject modelling approaches of EEG using different datasets obtained from 2 different EEG devices for machine learning purpose., the analysis and feature extraction of EEG signal for each trial has been done individually and the average responses were represented in the figures (Fig 3, Fig 4). Although it was a risk, we found data from central electrodes allowed reliable discrimination across subjects performing the tasks recorded using both devices.

We are happy

---

## [Editor Report · Decision Letter 1]

9 Jun 2022

Exploring EEG spectral and temporal dynamics underlying a hand grasp movement

PONE-D-21-37461R1

Dear Dr. Diwakar,

We’re pleased to inform you that your manuscript has been judged scientifically suitable for publication and will be formally accepted for publication once it meets all outstanding technical requirements.

Kind regards,

Mukesh Dhamala, Ph. D.

Academic Editor

PLOS ONE

---

## [Editor Report · Acceptance letter]

13 Jun 2022

PONE-D-21-37461R1 

Exploring EEG spectral and temporal dynamics underlying a hand grasp movement 

Dear Dr. Diwakar:

I'm pleased to inform you that your manuscript has been deemed suitable for publication in PLOS ONE. Congratulations! Your manuscript is now with our production department. 

Kind regards, 

on behalf of

Dr. Mukesh Dhamala 

Academic Editor

PLOS ONE